# Seismic Data Query Algorithm Based on Edge Computing

Tenglong Quan [1], Huifeng Zhang [1], Yonghao Yu [2,*], Yongwei Tang [2,3], Fushun Liu [1] and Hao Hao [2,*]

1 Shandong Earthquake Agency, Jinan 250014, China; 202185009009@sdu.edu.cn (T.Q.); 202185009008@sdu.edu.cn (H.Z.); 202115383@sdu.edu.cn (F.L.)
2 Shandong Computer Science Center (National Supercomputing Center in Jinan), Qilu University of Technology Shandong Academy of Sciences), Jinan 250353, China; tangyw@sdas.org
3 Key Laboratory of High Efficiency and Clean Mechanical Manufacture, School of Mechanical Engineering, Ministry of Education, Shandong University, Jinan 250061, China
* Correspondence: yuyh@sdas.org (Y.Y.); haoh@sdas.org (H.H.)

**Abstract:** Edge computing can reduce the transmission pressure of wireless networks in earthquakes by pushing computing functionalities to network edges and avoiding the data transmission to cloud servers. However, this also leads to the scattered storage of data content in each edge server, increasing the difficulty of content search. This paper investigates the seismic data query problem supported by edge computing. We first design a lookup mechanism based on bloom filter, which can quickly determine if there is the information that we need on a particular edge server. Then, the MEC-based data query problem is formulated as an optimization problem whose goal is to minimize the long-term average task delay with the constraints of computing capacity of edge servers. To reduce the complexity of problem, we further transform it as a Markov Decision Process by defining state space, action space and reward function. A novel DQN-based seismic data query algorithm is proposed to solve problem effectively. Extensive simulation-based testing shows that the proposed algorithm performances better when compared with two state-of-the-art solutions.

**Keywords:** edge computing; seismic content lookup; deep reinforcement learning

## 1. Introduction

Since the fifth generation of seismic devices, computer network technology has been introduced into seismic devices. A complete seismic telemetry acquisition system can be regarded as a relatively complex Local Area Network (LAN) system [1]. Seismic sensor network is essentially a special sensor distributed network, so it is necessary to use network research methods to study it, making the data transmission performance of the whole system optimal. In addition, current seismic devices have the characteristics of high density, large range and three-component acquisition. For wired seismic devices, simply increasing the number of instrument tracks will not only cause pressure on the bandwidth of the system, but also cause inconvenience to the construction. Wireless transmission has gradually become an important transmission method for seismic devices [2]. Due to the limited wireless bandwidth resources, if all the information generated by the seismic device is transmitted to the cloud server for unified management, it will waste a lot of communication resources.

In recent years, a new computing paradigm, mobile edge computing (MEC) [3] was proposed, which can move some of the computational functions to the network edge devices. In other words, we can perform distributed management of information on edge servers through MEC. In this way, the seismic device does not need to send all the information back to the cloud server for unified management, which can effectively reduce the transmission pressure [4]. However, this also introduces a new problem. Due to the lack of unified data management, it is difficult to quickly determine the location of data and content query also becomes a problem. So, fast data query in the wireless network of

earthquakes supported by edge computing has become an important problem. But due to the complexity of the problem and the limited computing resources of edge servers, there are still many challenges to be solved.

Deep reinforcement learning (DRL) solutions [5] help take appropriate actions according to the state of environment and have achieved great success in various kinds of control tasks, which can provide a new idea to solve the task allocation problem in wireless networks of earthquakes. Reinforcement learning has many advantages, among which optimization of long-term average reward through training on historical data and adaptability to the environment. One of the most remarkable algorithms in DRL is deep Q-learning (DQN) [6], which can deal with models in continuous state spaces.

In this paper, we consider an earthquake information searching system with multiple edge servers. Earthquake information is stored and distributed on edge servers. When we perform a search, we first need to determine location of the information. Considering multiple edge servers will store the information, we further need to determine which server can execute the task with the constraints of computing capacity and task delay. To solve the problem, we first propose a lookup mechanism to quickly determine the location of information by bloom filter. Then, we formulate the MEC-based data query problem as a Markov Decision Process (MDP) and propose a DRL algorithm to solve it. The main contributions of this paper are as follows:

- **Construct the System Model.** We propose a lookup mechanism by bloom filter, which can quickly determine if there is the information that we need on a particular edge server. In addition, we formulate the MEC-based data query as a long-term average optimization problem, whose goal is to optimize the service delay with the constraints of computing capacity.
- **DRL-based Algorithm.** Considering the complexity of problem, we further transform the problem into an MDP by defining the state space, action space and reward function. A model-free deep reinforcement learning algorithm is proposed to solve the problem. Instead of using a traditional $\epsilon$-greedy strategy, we introduce the confidence interval to explore action, which can improve the training efficiency of the model.
- **Comparison-based Evaluation.** We perform extensive simulations to evaluate the performance of our proposed algorithm. The simulation results show that our algorithm achieves better performance in comparison with two baselines.

The paper is organized as follows. The related works are reviewed in Section 2. The system model is introduced in Section 3. The algorithm is proposed in Section 4. Section 5 discusses the simulation results and Section 6 introduces limits of our paper and future work. Section 7 presents the conclusions.

## 2. Related Works

### 2.1. Edge Computing

As a new paradigm in computing and networking, edge computing is attracting extensive attention from academic and industry researchers. Some works have studied the computation offloading problem involving only a single edge node. For example, literature [7] focused on the energy consumption problem of computation offloading, and modeled it as a stochastic optimization problem with the goal of minimizing the energy consumption of task offloading while ensuring the average waiting queue length. The original problem was transformed by stochastic optimization method, and a dynamic energy-saving computation offloading method was proposed. Literature [8] formulated the task offloading problem, which joins uplink, downlink and computing resources allocation as a network of queues. The goal of optimization is to minimize the operational expenditure of computing resource providers. Then, authors transformed the problem into a Generalized Nash Equilibrium Problem and proposed two decentralized algorithms by introducing the penalty parameters to the coupling constraints. In [9], authors proposes an opportunistic access fog-cloud computing network to reduce latency and energy consumption of data transmission and computation. In this way, each mobile user can select

the fog node through opportunistic access method. With the constraints of users' quality of service requirements, they jointly optimized both resource allocation and computation offloading, and designed an iterative algorithm to solve the problem. However, there are always multiple edge nodes in a system. Due to the different number of terminal devices and the different amount of computing tasks in the service range of each edge node, the single edge node offloading model is easy to lead to heavy computing pressure on some edge nodes, while some edge nodes are relatively idle. This results in the load imbalance between edge nodes, which affects the service experience [10].

Some of the latest research works consider the cooperation of multiple edge nodes to complete computing tasks. The authors of [11] proposed a computation offloading optimization mechanism for multi-edge device cooperation, modeled the optimization problem as a mixed-integer nonlinear programming problem, and designed a preference-based two-sided matching algorithm to reduce the overall task execution delay and achieve load balancing between edge devices. Literature [12] classifies application tasks in terms of computation amount and communication cost to help the computation offloading decision, and proposes an offloading algorithm based on greedy task graph partitioning, which uses greedy optimization method to minimize task communication cost. Reference [13] considers the trade-off between task delay during computation offloading and energy consumption during wireless transmission, and proposes an efficient and distributed predictive offloading and resource allocation scheme for multi-layer fog computing system by predicting system traffic, which can significantly reduce task delay with only a small amount of prediction information value. The authors of [14] designed a software-defined fine-grained multi-access edge computing architecture, which co-managed network and computing resources, and designed a two-level resource allocation strategy based on deep reinforcement learning to provide effective computing offloading services. Data privacy is also an important issue in computing offloading. Combined federated learning, authors [15] proposed a distributed learning framework for computation offloading with the goal of minimizing the task delay and energy consumption. As shown in Table 1, these computation offloading methods select the offloading location from all the edge nodes. But in seismic data query, information is stored separately and not all edge nodes store the required information, so the traditional computing offloading method is difficult to apply to this scenario.

**Table 1.** Comparison of existing works.

| Literature | Cooperation of Nodes | Long-Term Average | Content Lookup |
|:---:|:---:|:---:|:---:|
| [7] | × | × | × |
| [8,9] | × | ✓ | × |
| [11,12] | ✓ | × | ✓ |
| [13–15] | ✓ | ✓ | × |
| our solution | ✓ | ✓ | ✓ |

*2.2. Wireless Network in Earthquakes*

With the rapid development of Internet of Things devices, wireless networks are more and more widely used in earthquake and meteorological observation. Authors [16] showed how IoT devices based on wireless network can share the information (i.e., slightest vibration detected by earthquake warning devices, water level, soil moisture) globally, and carried out an in-depth analysis of it. The authors of [17] introduced the construction of Nankai Trough Seafloor Observation Network for Earthquakes and Tsunami (N-net). The system is a hybrid network of wired and wireless. Due to the low cost and rapid deployment, wireless sensor networks can provide remote monitoring and sensing for many critical scenarios, such as earthquakes and floods. The authors of [18] proposed an architecture based on wireless sensor networks in smart city applications and used unmanned

aerial vehicles as offloading nodes. In [19], the authors introduced an open-source earthquake monitoring system, which was based on wireless network and energy-autonomous sensor nodes. Two emerging technologies in wireless network, LoRa and Message Queue Telemetry Transport (MQTT), was used in the system to reduce latency and packet delivery ratio. The authors of [20] designed an energy consumption scheduling scheme based on longitude and latitude coding algorithm and differential evolution algorithm to reduce overall energy consumption of wireless sensor network in the process of earthquake monitoring. In order to improve the utilization rate of network resources, they formed three-layer network architecture by building intermediate service layer to connect sensor nodes and sink nodes. Authors [21] proposed a low-cost, low-power and cloud-base seismic alert system. In order to process tasks based on computation and communication resources in the cloud, the system leveraged existing wireless networks to minimize costs and maximize communication speed. Although wireless networks have gradually become an important transmission method for seismic devices, there are few studies on seismic data query in wireless network.

*2.3. Quick Lookup Mechanism*

Focusing on lookup speed, memory consumption and false positive probability, the paper [22] proposed a smart mapping model named Pyramid-NN via neural networks to build an index algorithm, which was trained by real NDN names offline and preset in content routers in the future, that can not only reduce the memory consumption and the probability of false positive, but also ensure the performance of real NDN name lookup. To improve the performance of content lookup, the authors of [23] extended the cuckoo filter to integrate a Bloom filter, which is used to improve the performance of insertions. The algorithm, which was adjusted to the bucket size, can preserve the support for deletion of the original cuckoo filter without the additional memory accesses for lookup operations. To effectively remove false positives while completing all queries in a single memory access, the authors of [24] proposed an adaptation scheme based on one memory access bloom filter. The proposed algorithm is very suitable for the case of low memory bits per element. In [25], the authors observed that many exact-pattern-matching-intensive workloads can benefit from a DRAM-based processing-in-memory (PIM) architecture, and extended the model with several cost-effective modifications to improve the efficiency of content lookup. For the flow table lookup problem, the authors of [26] proposed a new flow table lookup scheme based on bloom filters and RAM, which offers a good compromise between cost and performance. The authors also verified the results by linearly searching the contents of secondary RAM to solve the problem of false positives of primary bloom filter.

The above mechanisms are suitable for scenarios where the information is only stored in one node. However, the required information is often stored in multiple edge nodes in a seismic data query, which means these lookup mechanisms cannot be directly applied to the information retrieval of wireless networks in earthquakes.

## 3. System Modeling

This section proposes the system model for task offloading. The scenario is described first and the problem is presented next. Table 2 lists the mathematical notations.

**Table 2.** Mathematical notations.

| Notation | Explanation |
| --- | --- |
| $\mathcal{N}$ | Set of edge servers |
| $F^n$ | Computing capacity of edge server $n$ |
| $\mathcal{K}$ | Set of tasks |
| $\mathcal{T}$ | Set of time slots |
| $c_k$ | Computing resource requirement of task $k$ |

**Table 2.** *Cont.*

| Notation | Explanation |
|---|---|
| $o_k$ | Data size of task $k$ |
| $p_n$ | Fixed transmission power of $n$ |
| $B_{m,n}$ | The channel bandwidth between $n$ and $m$ |
| $h_{m,n}$ | Channel gain between $n$ and $m$ |
| $\sigma^2_{m,n}$ | Noise power |
| $T_n^k$ | Service delay status of edge server $n$ for task $k$ |
| $D^k$ | Edge server status for task $k$ |
| $l$ | Size of the bit array |
| $j$ | The number of element |
| $p$ | The acceptable misjudgment rate |
| $i$ | The number of hash functions |
| $tc_m^k$ | Computation delay of task $k$ |
| $r_{m,n}$ | Transmission rate between edge server $m$ and $n$ |
| $tt_{m,n}^k$ | Transmission delay of service $k$ |
| $f_m^k$ | Computation resources that $m$ assigns to $k$ |

### 3.1. Scenario Description

An earthquake information searching system with multiple edge servers is considered, as shown in Figure 1. There are three layers: cloud server, edge servers and seismic devices. In reality, seismic devices collect various types of information, then they will transmit the contents which are collected by them to the related edge server due to the limited computing and storage resources of seismic devices. The edge server stores the information uploaded by the seismic device locally after receiving it. If the edge service runs out of store space, it will also offload some information to the cloud server. The set of edge server is denoted as $\mathcal{N} = \{1, 2, \ldots, N\}$. Each edge server $n$ is equipped with computing capacity $F^n$ (e.g., the maximum frequency of CPU) to support the information searching. We use task $k$ to denote the task of searching an element, and the set is denoted as $\mathcal{K} \triangleq \{1, 2, \ldots, K\}$. For task $k$, there are two important attributes $(c_k, o_k)$, where $c_k$ denotes the total computation resources required to finish the task $k$, i.e., the number of CPU cycles, and $o_k$ is the data size of the element that needs to be transmitted. Without loss of generality, we assume the time is slotted [27], i.e., $\mathcal{T} = \{1, 2, \ldots, T\}$.

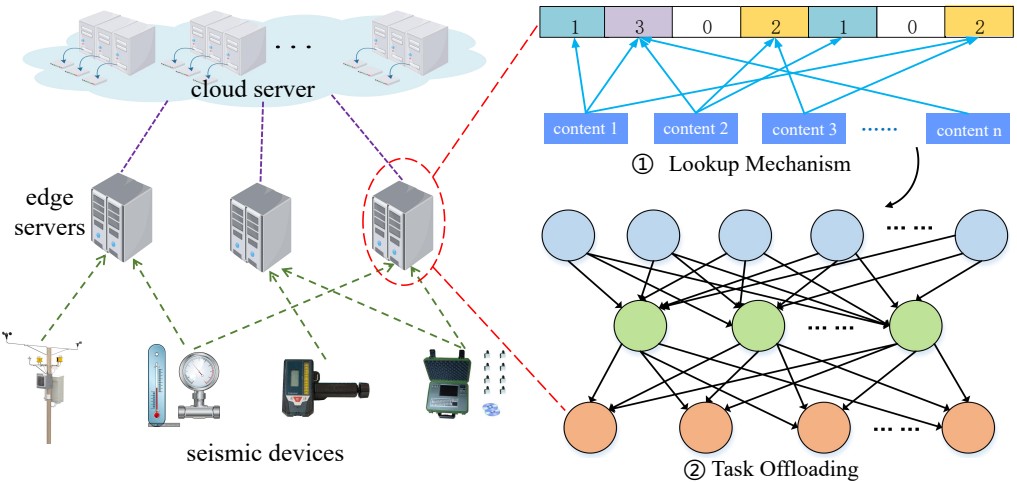

**Figure 1.** System architecture.

There are two phases for a task. First, we need to find the position of the element. Different edge server stores different information. If we search the whole network, it will not only increase the pressure of network traffic, but also bring a huge amount of search computation to the system. So, we need to filtering positions of the element quickly and accurately to avoid a global lookup. There may be multiple edge servers storing related elements after filtering. Then, we need to further determine which edge server to obtain the element from according to the service delay. In the following subsections, we will introduce the lookup mechanism and the calculation of service delay.

### 3.2. Lookup Mechanism

When looking for a certain piece of information, we first need to determine which edge server the information is located in. We propose the quick lookup algorithm based on bloom filter (BF) to make quick judgments. Bloom filter is a space-efficient random data structure, which can be seen as an extension of bit-map [28]. It is designed as a bit array with *j* elements, and initially all the bits are set to 0. When an element is added to the set, *i* hash functions are used to map the element to *i* points in a bit array and set them to add 1. When we retrieve it, we just check to see if all the points are not zero and we will (approximately) know if it is in the set.

The main process of our lookup algorithm based on bloom filter is as follows. We first construct a bloom filter for each edge server, and all bloom filters are shared across all servers. When an element lookup is performed, we first process the element with different hash functions, and each hash function returns an integer as the index value of the bit array. Then, we will match all bloom filters. If all the indexed values are not zero, it means the element is in the bloom filter and we can send an element request to that edge server. Otherwise, the element is not in the edge server. By the way, the bloom filter also has a certain probability of misjudgment. The size of the bit array (denoted by *l*) is very important. If it is too small, all the bits will quickly be assigned the value, increasing the chance of false positives. The calculation of *l* is as follows.

$$l = -\frac{j \cdot lnp}{(ln2)^2} \tag{1}$$

where *ln* is the logarithm of the natural logarithm with the constant e as the base, *j* is the number of element, and *p* is the acceptable misjudgment rate.

Besides, the number of hash functions (denoted by *i*) is also important for an even distribution of index values. The calculation of *i* is as follows.

$$i = \frac{l}{j}ln2 \tag{2}$$

When an edge service needs to add a new element, we use the *i* hash function to map the element into *i* index values. Then, we increment the value of the bit array at this index by 1. If an element is removed from a seismic service, we first also obtain the *i* index values, and then decrement the value of the bit array at this index by 1.

### 3.3. Service Delay

Since there may be multiple edge servers storing the element, the bloom filter will target multiple servers. So, we have to determine the location which can provide fastest search speed. If edge server *n* needs to search an element which can be denoted as task *k*, and edge server *m* is a target which is selected by bloom filter, the computation delay is defined as:

$$tc_m^k = c_k / f_m^k \tag{3}$$

where $f_m^k$ is the computation resources that BS *m* assigns to service *k* per second (cycles/s).

When the element is found, edge server $m$ need to transmit the element back to device $n$. The transmission rate is:

$$r_{m,n} = B_{m,n} log_2(1 + \frac{p_m h_{m,n}}{\sigma_{m,n}^2}) \tag{4}$$

where $B_{m,n}$ is the channel bandwidth between $n$ and edge server $m$, $p_m$ is the fixed transmission power of edge server $m$, $h_{m,n}$ is the channel gain, and $\sigma_{m,n}^2$ is the noise power. The transmission delay of service $k$ is

$$tt_{m,n}^k = o_k / r_{m,n} \tag{5}$$

The total service delay that $n$ requests an element $k$ from edge server $m$ is

$$T_{m,n}^k = tc_m^k + tt_{m,n}^k \tag{6}$$

### 3.4. Problem Formulation

If edge server $n$ needs to search task $k$ in time slot $t$, we use variables $x_{n,k}^t$ to denote the final query location. In order to optimize the system query efficiency, we formulate the earthquake data query problem as follows:

$$\min_{x_{n,k}^t} \lim_{T \to \infty} \frac{1}{T} \sum_{n \in \mathcal{N}} \sum_{k \in \mathcal{K}} T_{x_{n,k}^t,n}^k$$
$$s.t. \sum_{k' \in \mathcal{K}} f_{x_{n,k}^t}^{k'} \leq F^{x_{n,k}^t}, \qquad \forall x_{n,k}^t \in \mathcal{N} \tag{7}$$

where the goal is the long-term average service delay, and the constraint is the limited computing capacity of edge servers.

It is difficult to solve the optimization problem. The goal of problem is the long-term average service delay, which always need future information if we use traditional approach (i.e., dynamic programming) to solve. However, it is impractical to obtain complete future information in a dynamic network [29]. In addition, the state transition probability of the optimization problem is also difficult to provide due to the unpredictability of requests. So we cannot solve the problem by traditional methods.

## 4. Algorithm Design

Deep reinforcement learning is an efficient way to solve the problem with the goal of long-term average optimization. DRL can use data sampling to replace the state transition probability, so that it can solve the optimization problem without complete future information. In this section, we first transform the optimization problem into an MDP, and then propose an algorithm based on DQN to solve it.

### 4.1. Markov Decision Process

In this subsection, we transform the problem into an MDP by defining state space, action space, and reward function.

***The State Space***: System state is a description of the current system environment. In this problem, the state is made up of service delay status and edge server status. We denote the service delay status of edge server $n$ for task $k$ as a vector $T_n^k = [T_{1,n}^k, T_{2,n}^k, \ldots, T_{N,n}^k]$. The edge server status for task $k$ is denoted as a vector $D^k = [d_1^k, d_2^k, \ldots, d_N^k]$, where $d_n^k = 1$ if edge server $n$ has the related element of task $k$, else $d_n^k = 0$. So, the state space of edge server $n$ is given as

$$s_n = (T_n^k, D^k), \quad where \ T_n^k \in \mathcal{N}, D^k \in \{0, 1\} \tag{8}$$

***The Action Space***: In the problem, we need to determine the element position which also the action of MDP. We use $a_n$ to denote the action, where $a_n = 1$ means that we can research the element from edge server $n$, and $a_n = 0$ means we will not obtain the element from edge server $n$. So the size of action space is the number of edge servers $N$. Additionally, there may be some actions that can not satisfy the constraints of computing capacity, which are called as illegal actions.

***The Reward Function***: The reward function defines the immediate value of taking action $a$ in state $s$. As our goal is to minimize the service delay, the action which can bring small service delay will obtain a large reward. We define the reward as the inverse of service delay. Additionally, illegal actions should be avoided, so the reward is defined as a penalty $Pu$, which is a negative number. Based on the above principles, we define the reward function as follows:

$$r_k^t = \begin{cases} Pu, & if\ a_m\ is\ illegal \\ 100/(100 + T_{m,n}^k), & if\ a_m\ is\ legal \end{cases} \tag{9}$$

The first component is for illegal actions and the second is for legal actions. If we use the inverse of service delay to denote the reward directly, it will cause the reward function to fluctuate too much when the service delay is small. For example, if service delay $T_{m,n}^k = 0.1$, the reward is 10. If $T_{m,n}^k = 0.2$, the reward is 5. The service delay is increased by 0.1, but the reward function is decreased by 5. To avoid this phenomenon, we added a certain cardinality to set the reward function instead of using the inverse directly.

*4.2. DRL-Based Algorithm*

The DQN is a value-based policy algorithm in reinforcement learning. Its purpose is to approximate a Q-value function that evaluates the value of each action by neural network. $Q(s, a)$ is the expectation that an action $a$ can be obtained in the state $s$. The environment will feedback the corresponding reward according to the action of the agent $r$. So the goal of DQN is to train a neural network to estimate the Q-value.

DQN has two tools, experience replay and fixed Q-targets. Because there is a relationship between two consecutive action state samples, but the neural network belongs to a nonlinear model, it must ensure that the samples are independent and identically distributed, so the off-line learning method and the replay buffer are used. The experiences are randomly drawn in the replay buffer, which can effectively shuffle the correlations between the samples. Second, another way to eliminate the correlation is to make duplicate neural networks, one to obtain the estimated Q and another to obtain the actual Q, with the second network replicating the parameters of the first network every certain number of steps. In general, the two networks have the same structure but different parameters. These are Fixed Q-targets.

There are also many other deep reinforcement learning algorithms (such as A2C, A3C, DDPG). However, the problems targeted by these algorithms themselves often have continuous action spaces. The action space proposed in this paper is discrete. Although the three algorithms mentioned above can be changed into discrete action space algorithms, their accuracy will be affected. After comprehensive consideration, DQN is selected as the reinforcement learning algorithm to solve the optimization problem to support the discrete action space.

We propose a solution algorithm based on DQN, using $Q(s, a)$ representing the action $a$ taken in the state $s$. The traditional $\epsilon$-greedy strategy will always choose the non-optimal action with a probability $\epsilon$, even if the model has converged, which causes the wasting of resources. To avoid the above problem, we introduce the Upper Confidence Bounds (UCB) to design the exploring strategy. For state $s$ and action $a$, the UCB is defined as follows:

$$UCB(s, a) = Q(s, a) + \sqrt{\frac{2lnu}{u_j}} \tag{10}$$

where $u$ is the total training times, and $u_j$ is the times that an action $a$ is selected in the state $s$. The action which has the largest $UCB(s, a)$ is selected at each time. With the training of model, the probability of the action that is selected less times will be executed with larger probability, which ensures the effectiveness of exploration. The model is trained by gradient descent, and the loss function is the mean square error between the target value and the output value of the network. The definition of the loss function is shown as

$$loss = \frac{[Q_{target}(s^t, a^t) - Q(s^t, a^t)]^2}{2} \tag{11}$$

where the $Q_{target}(s, a)$ is defined as

$$Q_{target}(s^t, a^t) = r^t + \gamma \max_{a'} Q(s^{t+1}, a^t) \tag{12}$$

where $\gamma$ is the discount factor. Our algorithm is shown as Algorithm 1.

---

**Algorithm 1** DQN-based seismic data query algorithm.

---

1: Select edge servers by lookup mechanism;
2: Initialize the replay memory;
3: Initialize the main network parameter $\theta$ and Q-values $Q(s, a)$ randomly;
4: Initialize target network parameter $\theta' = \theta$ and Q-value $Q'(s, a) = Q(s, a)$;
5: **for** $i = 1, \dots, P$ **do**
6:    Initialize $s^1$ as first state;
7:   **for** $t = 1, \dots, T$ **do**
8:      Input $s^t$ and select action $a^t$ by $UCB(s, a)$ exploration strategy;
9:      Execute action $a^t$, get the new state $s^{t+1}$ and reward $r^t$ ;
10:     Store four-tuple $(s^t, a^t, r^t, s^{t+1})$ in replay memory;
11:     Sample a mini-batch samples $(s^j, a^j, r^j, s^{j+1})$ from replay memory randomly;
12:     Get the loss function by Equation (11) ;
13:     Perform gradient descent with respect to the network parameters $\theta$ by loss function;
14:     **if** $t\%L == 0$ **then**
15:       Update target network parameter $\theta' = \theta$
16:     **end if**
17:   **end for**
18: **end for**

---

The algorithm first select edge servers that contain the element via a lookup mechanism. Then, we use a DRL-based algorithm to determine the final edge server to query. We initialize the replay memory and network parameters. Then, algorithm runs multiple loops. In each loop, the algorithm select the edge server which can make the $UCB(s, a)$ largest, then performance action, obtain the reward and the next state, and store the four-tuples $(s^t, a^t, r^t, s^{t+1})$ in replay memory. Finally, select a mini-batch from replay memory, and train the model with gradient descent by loss function as lines 12–16.

## 5. Simulation Results

In this section, we conduct a large number of simulation experiments to verify the performance of our algorithm. We analyze the performance in term of convergence, task delay, computing capacity and number of edge servers.

### 5.1. Simulation Scenario and Setup

A network scenario with 30 seismic devices, 6 edge nodes, and 1 remote cloud server is considered in the experiment. The computing capacity of the edge node ranges from [30, 40] GHz, and the transmission rate between edge servers ranges from [10, 15] Mbps. The value range of the amount of data transmitted in a computing task is [1, 2] Mbits. The value range of the computing workload in a computing task is [0.5, 1.5] Gigacycles [30].

User request data are generated through simulation. We assume the request frequency conforms to the Zipf distribution with parameter $\lambda = 1$. The average number of requests to service $k$ is as follows:

$$\bar{D}(k) = \frac{V}{r(k)^{\lambda}} \cdot N \tag{13}$$

where $r(k)$ is the ranking of request frequency for service $k$, $N = 30$ is the number of seismic devices and $V = 0.1$. Table 3 lists the parameters for this experiment. During neural network training, the batch size is set to 16, the learning rate to $10^{-3}$, and the discount factor to 0.9. At the same time, the stochastic gradient descent (SGD) optimizer is used to perform training optimization of the model. In order to evaluate performance improvements, we compare our algorithms with the following solutions.

- **Flooding method**: When the edge server receives the lookup request, if there is no relevant content after the local lookup, it will forward the request to all other edge server nodes. When the other edge server receives the request, it will perform the lookup of the task.
- **CFBF-based method**: We apply Cuckoo Filters With an Integrated Bloom Filter (CFBF) [23] to the content lookup of edge servers. When the edge server receives the lookup request, it will quickly determine which edge servers store the content by cuckoo filters and randomly select a edge server to forward the request.

We compare our algorithm with two solutions: Flooding method and CFBF-based method. The differences between the three algorithms are as shown in Table 4. Flooding method does not have a lookup mechanism, and the task offloading method only forwards the request to all other edge server nodes. CFBF-based method quickly locates content by CFBF, but does not combine with the task offloading method. We quickly locate content via BF and design a DQN-based task offloading method to finally determine the location of information in our algorithm.

**Table 3.** Parameter settings for simulations.

| Parameters | Value |
|---|---|
| The number of seismic devices | 30 |
| The number of edge servers | 6 |
| Computing capacity of edge server $F^n$ | [30, 40] GHz |
| Transmission rate of the seismic device to the edge server | [10 Mbps, 15 Mbps] |
| Computing workload of a task | [0.5, 1.5] Gigacycles |
| The amount of data transmitted of a computing task | [1 Mbits, 2 Mbits] |
| Batch size of neural network | 16 |
| Learning rate of neural network | $10^{-3}$ |
| optimizer | SGD |

**Table 4.** Comparison of algorithms.

| Algorithm | Lookup Mechanism | Task Offloading Method |
|---|---|---|
| Flooding method | None | None |
| CFBF-based method | CFBF | None |
| Our method | BF | DQN-based |

*5.2. Performance Evaluation*

Figure 2 shows the convergence behaviors with different learning rate $\alpha$ in the training phase. When $\alpha = 10^{-1}$, neural network can converge quickly. It will converge after about

180 episodes, but the model fluctuates greatly. This is because the model learning rate is too large and it is difficult to converge to the optimal point, which leads to the oscillation of model. When $\alpha = 10^{-2}$, the neural network can converge quickly, which can converge after about 230 episodes, but the model still fluctuates greatly. When $\alpha = 10^{-3}$, although the neural network converges relatively slow, up to 400 episodes, the convergence of the neural network model is very good. Therefore, we set the learning rate as $\alpha = 10^{-3}$ in the training phase of the model.

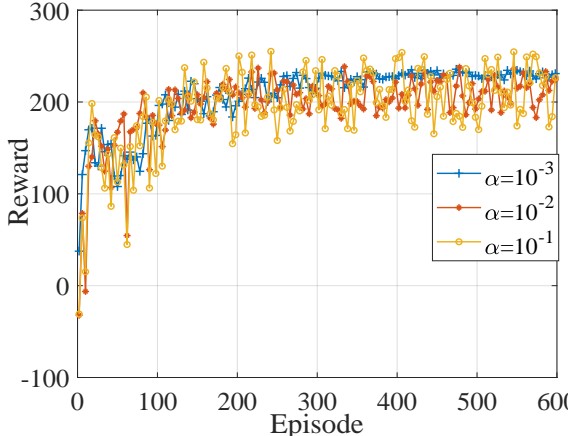

**Figure 2.** Convergence behavior with different learning rates.

Figure 3 shows the loss value with different batch size in the training phase, which is the number of four tuples selected in each episode from replay memory. As shown in the figure, when the batch size is 4, the convergence speed of the model is slow and the model fluctuates largely. It converges about 300 episodes. The reason is that the smaller the batch size, the more accidental the data will be. When the batch size is increased from 4 to 16, the convergence speed of the model is significantly accelerated and the model is more stable. It converges about 400 episodes. Although the training time and calculation amount of each round are positively related to the batch size, that is, the larger the batch size, the higher number of four tuples need to be trained in each episode and the longer the training time, we choose the batch size to be 16 to improve the convergence of the algorithm.

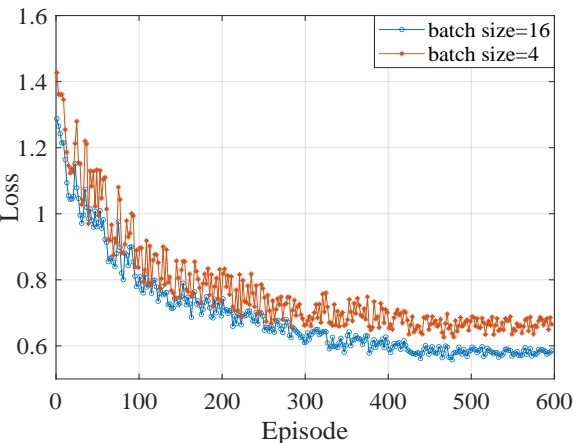

**Figure 3.** Loss value with different batch sizes.

Figure 4 shows the task delay (lookup time) of three algorithms. We find that the task delay of the three algorithms is very low at the beginning of experiment. The task delay of flooding method is stable at 200 ms, and the task delay of CFBF-based is 180 ms. Compared with these two algorithms, our algorithm can reduce the delay by about 20% and 11.1%, which is up to 160 ms. The reason is that all edge servers are idle when the system starts,

so edge servers have enough computing resource to complete task, which leads to the low delay. As the system runs and the number of tasks increases, the service delay will gradually rise and become stable. Compared with the other two algorithms, our algorithm has the lowest task delay due to BF and the choice of computing location, which means our algorithm can reduce the delay of seismic data query without increasing network overhead. On the other hand, flooding method has to forward tasks to all other edge servers, which consumes a lot of computational resources, so it has the worst performance.

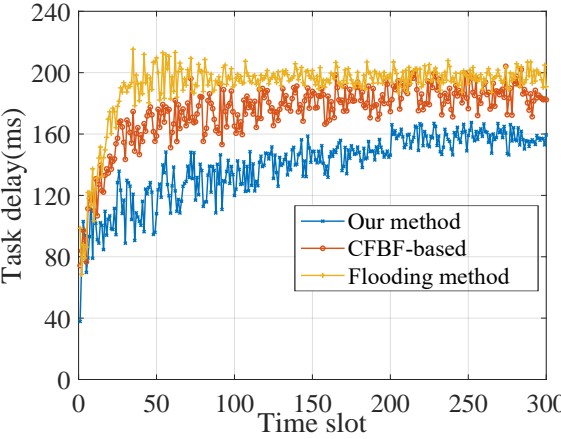

**Figure 4.** Task delay of three algorithms.

Figure 5 shows the effect of computing capacity of edge servers. It is clear that as the computing power of the edge server increases, the task delay decreases. It is easy to explain that the edge server has sufficient computing resources to perform multiple tasks at the same time, thus reducing the average task delay. Therefore, in practice, if we can improve the computing capacity of edge servers, we can also effectively reduce the delay of tasks. Overall, our algorithm has the lowest task delay and best performance. It is worth noting that CFBF-based algorithm has a better performance than our algorithm when the computing capacity of edge server is 3 GHz. The reason is that the execution cost of our algorithm will be slightly higher than that of CFBF-based algorithm due to the use of BF and DRL. So when computing resources are tight, the benefit brought by our algorithm is lower than the overhead when the algorithm is executed, resulting in slightly worse performance.

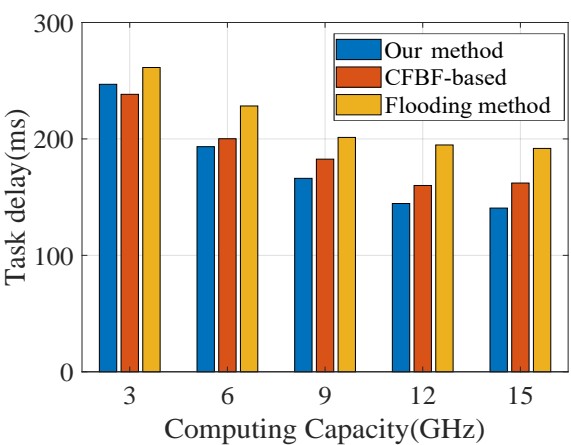

**Figure 5.** The effect of computing capacity.

Figure 6 shows the effect of number of edge servers. We find that as the number of edge servers increases, the average task delay of our algorithm and CFBF-based algorithm decreases instead. As the number of edge servers increases, the number of tasks performed on each edge server decreases, the computational pressure on the edge servers decreases,

and consequently the task delay decreases. But for flooding method, the average task delay does not decrease as the number of edge servers increases, because flooding method need to forward the task to every edge server, and the number of tasks performed on each edge server does not decrease as the number of edge servers increases. On the contrary, due to the increase in system complexity, the service delay increases slightly.

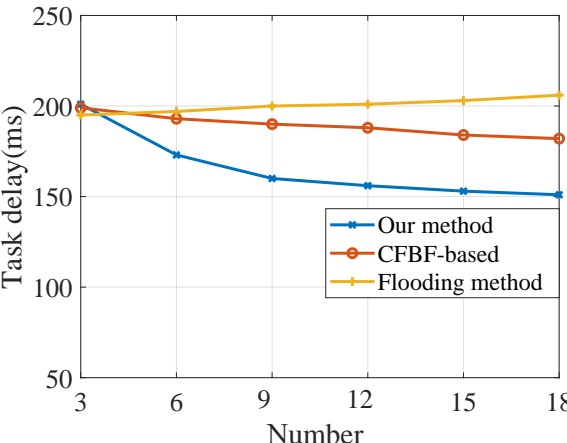

**Figure 6.** The effect of the number of edge servers.

## 6. Discussion

We proposed a seismic data query algorithm based on edge computing in this paper. There are also some limitations of our algorithms. First, we did not consider the error rate of the bloom filter, which related to the size of the bit array and the number of elements. When an error occurs, we need to make decisions again, which may incur additional overhead. Second, mobility is not considered in the problem. User mobility will increase the complexity of the problem significantly. For future work, we will further consider user mobility and analyze the impact of the error rate of bloom filter, in order to better improve the efficiency of seismic data query.

## 7. Conclusions

In this paper, we focused on the seismic data query problem, which is based on edge computing technology in an earthquakes wireless network. First, a BF-based lookup mechanism is proposed to quickly determine the location of content. In order to obtain the content, we need to further determine which edge server the content query task will be offloaded to. We formulated the problem as an Markov Decision Process by defining the state space, action space and reward function. Considering the complexity of the problem, a DQN-based algorithm which can effectively solve the problem was proposed. Finally, the extensive simulation-based performance evaluation showed that our proposed solutions had better performance than the other two baseline algorithms.

**Author Contributions:** Conceptualization—methodology, T.Q.; validation—investigation, H.Z.; formal analysis—data curation, Y.Y.; review—editing, Y.T.; visualization, F.L.; writing—original draft preparation, H.H. All authors have read and agreed to the published version of the manuscript.

**Funding:** This work is supported by the National Natural Science Foundation of Shandong Province under Grant ZR2022QF040; the QLU Pilot Project of Integration of Science, Education and Production under Grant 2022PX083.

**Institutional Review Board Statement:** Not applicable.

**Informed Consent Statement:** Not applicable.

**Data Availability Statement:** Not applicable.

**Conflicts of Interest:** The authors declare that there is no conflict of interest regarding the publication of this paper.

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
