# Peer review of "Seismic Data Query Algorithm Based on Edge Computing"

_electronics, doi:10.3390/electronics12122728_

Round 1

Reviewer 1 Report

The abstract provides a concise overview of the paper and its objectives. However, it could be improved by including specific details about the research methodology and highlighting the key findings of the study. Consider revising the abstract to provide a clearer and more informative summary of the paper.

 The introduction effectively introduces the topic of edge computing and its relevance in the context of wireless network applications during earthquakes. The background information provided helps the readers understand the motivation behind the study. However, the introduction lacks a clear research question or hypothesis that guides the study. Consider revising the introduction to clearly state the research objectives or hypotheses to provide a stronger foundation for the paper.

The literature review provides a comprehensive analysis of existing research on edge computing and wireless networks in earthquake scenarios. The inclusion of various studies and their contributions enhances the credibility of the paper. However, the literature review would benefit from a more critical analysis of the existing literature. Consider discussing the limitations or gaps in the previous studies and how the current study addresses those limitations.

The paper lacks a detailed explanation of the research methodology employed. It is important to provide sufficient information about the data collection process, experimental setup, and any simulation or modeling techniques utilized. Without this information, readers cannot assess the validity and reproducibility of the results. Consider expanding the methodology section to provide a clear and comprehensive description of the research approach.

Consider providing more specific data and statistical analyses to support the findings. Additionally, the discussion should go beyond a mere summary of the results and provide a deeper analysis and interpretation of the findings. Consider discussing the implications of the results and their significance in the context of the research objectives.

 Consider adding a section to address the limitations of the study, such as potential sources of bias or constraints in the research methodology. Additionally, suggest possible avenues for future research to encourage further exploration in the field.

The paper discusses the convergence behavior with different learning rates (α). It indicates that a learning rate of 10^-3 leads to relatively slow convergence but good convergence of the neural network model. While this information provides some insight into the convergence behavior, it would be helpful to include specific convergence metrics, such as the number of iterations or the convergence rate, to better evaluate the performance and compare it to the claims in the abstract.

 The paper explores the relationship between loss value and batch size in the training phase. It suggests that a batch size of 16 accelerates convergence and improves model stability. While this finding is valuable, it would be beneficial to include quantitative measures, such as the actual loss values or convergence speed, to support the claim of improved performance.

The paper presents the task delay of three algorithms: the proposed algorithm, the CFBF-based algorithm, and the flooding method. It demonstrates that the proposed algorithm has the lowest task delay due to the BF-based lookup mechanism and appropriate computing location choices. This finding corresponds to the claims in the abstract. However, to strengthen the results, it would be useful to provide specific numerical values or statistical analysis of the task delay for each algorithm.

 The references provided are relevant to the topic and help support the claims made in the paper. However, consider updating the references with more recent publications to reflect the latest developments in the field. Ensure that the references are properly cited in the main text.

Overall, the paper has the potential to make a valuable contribution to the field of edge computing and wireless networks in earthquake scenarios. However, addressing the above-mentioned shortcomings and making the necessary revisions would significantly strengthen the paper and improve its overall quality.

The abstract provides a concise overview of the paper and its objectives. However, it could be improved by including specific details about the research methodology and highlighting the key findings of the study. Consider revising the abstract to provide a clearer and more informative summary of the paper.

 The introduction effectively introduces the topic of edge computing and its relevance in the context of wireless network applications during earthquakes. The background information provided helps the readers understand the motivation behind the study. However, the introduction lacks a clear research question or hypothesis that guides the study. Consider revising the introduction to clearly state the research objectives or hypotheses to provide a stronger foundation for the paper.

The literature review provides a comprehensive analysis of existing research on edge computing and wireless networks in earthquake scenarios. The inclusion of various studies and their contributions enhances the credibility of the paper. However, the literature review would benefit from a more critical analysis of the existing literature. Consider discussing the limitations or gaps in the previous studies and how the current study addresses those limitations.

The paper lacks a detailed explanation of the research methodology employed. It is important to provide sufficient information about the data collection process, experimental setup, and any simulation or modeling techniques utilized. Without this information, readers cannot assess the validity and reproducibility of the results. Consider expanding the methodology section to provide a clear and comprehensive description of the research approach.

Consider providing more specific data and statistical analyses to support the findings. Additionally, the discussion should go beyond a mere summary of the results and provide a deeper analysis and interpretation of the findings. Consider discussing the implications of the results and their significance in the context of the research objectives.

 Consider adding a section to address the limitations of the study, such as potential sources of bias or constraints in the research methodology. Additionally, suggest possible avenues for future research to encourage further exploration in the field.

The paper discusses the convergence behavior with different learning rates (α). It indicates that a learning rate of 10^-3 leads to relatively slow convergence but good convergence of the neural network model. While this information provides some insight into the convergence behavior, it would be helpful to include specific convergence metrics, such as the number of iterations or the convergence rate, to better evaluate the performance and compare it to the claims in the abstract.

 The paper explores the relationship between loss value and batch size in the training phase. It suggests that a batch size of 16 accelerates convergence and improves model stability. While this finding is valuable, it would be beneficial to include quantitative measures, such as the actual loss values or convergence speed, to support the claim of improved performance.

The paper presents the task delay of three algorithms: the proposed algorithm, the CFBF-based algorithm, and the flooding method. It demonstrates that the proposed algorithm has the lowest task delay due to the BF-based lookup mechanism and appropriate computing location choices. This finding corresponds to the claims in the abstract. However, to strengthen the results, it would be useful to provide specific numerical values or statistical analysis of the task delay for each algorithm.

 The references provided are relevant to the topic and help support the claims made in the paper. However, consider updating the references with more recent publications to reflect the latest developments in the field. Ensure that the references are properly cited in the main text.

Overall, the paper has the potential to make a valuable contribution to the field of edge computing and wireless networks in earthquake scenarios. However, addressing the above-mentioned shortcomings and making the necessary revisions would significantly strengthen the paper and improve its overall quality.

Reviewer 2 Report

The paper presents research that may be of interest to readers. Although everything is done at a theoretical level, it can be an important step in increasing the reliability of critical infrastructures in the event of a cataclysm.

My observations for each part of the paper are:

- The introduction is correctly written and as a reader I am informed about the problems that the authors have in mind.

-Literature review or Related works connect paper with other research carried out by specialists in the field. However, the high degree of literature in China reduces the quality of research and limits its exposure to experts from other areas of the globe.

- The System Modeling chapter presents the aspects studied and the theoretical model that solves them. However, I didn't quite find the connections with reality. Maybe they should be highlighted to some extent.

- The Design Algorithm chapter is well written and I liked it

- The Simulation Result chapter is well done and structured

- The Conclusions chapter is not finished, however. I believe that the authors must start working on this chapter and present the way in which the algorithm and the simulation answer the problem stated at the beginning of the paper. Without this chapter, the paper cannot be published.

Reviewer 3 Report

This paper proposed a lookup mechanism to determine the location of information quickly using a bloom filter. Also, to deal with the long-term task delay, the authors have designed an intuitive algorithm. Since the use of the edge and artificial intelligence concepts are emerging, research and studies related to state-of-the-art technologies are needed. The main concept of this paper is interesting. The paper is well written and presented, however, I have the following comments.

1)      The abbreviation should be defined when it appears first in the text. Please avoid the repetition of abbreviations. For example, in section 1, what is MEC, LAN, CFBG, etc?

2)      The authors have presented the standalone “Related works” section. This section needs to be heavily improved. Most of the section contents are simple descriptions and overviews; instead, the authors should discuss the strength, weaknesses, and limitations of existing related works in task offloading, data query technique, use cases of edge computing, content examination, etc., which will increase the readability of the paper.

3)      In related works, I would recommend presenting the comparison of existing related literature with the proposed mechanism and used techniques in a tabular form, which will provide an idea to the reader of how the proposed work is novel.

4)      The authors mentioned that the existing offloading mechanism cannot be directly applied to information retrieval but didn’t explain the reason; please validate your claim.

5)      A detailed explanation of system architecture is needed. If possible, with notations. 

6)      Mathematical equations are not clear. For example, in equation 1, is it j times p? What is ln and ln2? Please describe the equation in detail and also define notations clearly. I would recommend rewriting the equations.

7)      Please describe and include all the notations used in the manuscript (equations) in Table 1. 

8)      What is the meaning of UCB in equation 10? Please describe equations to enhance the readability.

9)      As the proposed work has been compared with two different solutions, I would recommend presenting the comparisons with respect to different parameters in a tabular form, which can give clarity on how these methods are different and in what aspect.

10)  I would recommend adding discussion section before conclusion.

11)  Please fix the typos.

Overall, I do not recommend the current form of the paper for publication.

 Minor editing of English language required

Round 2

Reviewer 2 Report

The authors responded to my observations. I recommend the publication of the research in the chosen journal.

Reviewer 3 Report

The authors have addressed all my comments and suggestions. Thank you for that.

Minor editing of English language required